# Transgelin, a p53 and PTEN-Upregulated Gene, Inhibits the Cell Proliferation and Invasion of Human Bladder Carcinoma Cells In Vitro and In Vivo

**DOI:** 10.3390/ijms20194946

**Published:** 2019-10-07

**Authors:** Ke-Hung Tsui, Yu-Hsiang Lin, Kang-Shuo Chang, Chen-Pang Hou, Pin-Jung Chen, Tsui-Hsia Feng, Horng-Heng Juang

**Affiliations:** 1Department of Urology, Chang Gung Memorial Hospital-Linkou, Kwei-Shan, Tao-Yuan 33302, Taiwan; t2130@cgmh.org.tw (K.-H.T.); linyh@doctorvoice.org (Y.-H.L.); glucose1979@gmail.com (C.-P.H.); 2Graduate Institute of Clinical Medical Science, College of Medicine, Chang Gung University, Kwei-Shan, Tao-Yuan 33302, Taiwan; 3Department of Anatomy, College of Medicine, Chang Gung University, Kwei-Shan, Tao-Yuan 33302, Taiwan; D0501301@stmail.cgu.edu.tw (K.-S.C.); lulu581623@gmail.com (P.-J.C.); 4Graduate Institute of Biomedical Sciences, College of Medicine, Chang Gung University, Kwei-Shan, Tao-Yuan 33302, Taiwan; 5School of Nursing, College of Medicine, Chang Gung University, Kwei-Shan, Tao-Yuan 33302, Taiwan; thf@mail.cgu.edu.tw

**Keywords:** bladder, TAGLN, F-actin, PTEN, p53, tumorigenesis, proliferation, invasion

## Abstract

Transgelin (TAGLN/SM22-α) is a regulator of the actin cytoskeleton, affecting the survival, migration, and apoptosis of various cancer cells divergently; however, the roles of TAGLN in bladder carcinoma cells remain inconclusive. We compared expressions of TAGLN in human bladder carcinoma cells to the normal human bladder tissues to determine the potential biological functions and regulatory mechanisms of TAGLN in bladder carcinoma cells. Results of RT-qPCR and immunoblot assays indicated that TAGLN expressions were higher in bladder smooth muscle cells, fibroblast cells, and normal epithelial cells than in carcinoma cells (RT-4, HT1376, TSGH-8301, and T24) in vitro. Besides, the results of RT-qPCR revealed that TAGLN expressions were higher in normal tissues than the paired tumor tissues. In vitro, TAGLN knockdown enhanced cell proliferation and invasion, while overexpression of TAGLN had the inverse effects in bladder carcinoma cells. Meanwhile, ectopic overexpression of TAGLN attenuated tumorigenesis in vivo. Immunofluorescence and immunoblot assays showed that TAGLN was predominantly in the cytosol and colocalized with F-actin. Ectopic overexpression of either p53 or PTEN induced TAGLN expression, while p53 knockdown downregulated TAGLN expression in bladder carcinoma cells. Our results indicate that TAGLN is a p53 and PTEN-upregulated gene, expressing higher levels in normal bladder epithelial cells than carcinoma cells. Further, TAGLN inhibited cell proliferation and invasion in vitro and blocked tumorigenesis in vivo. Collectively, it can be concluded that TAGLN is an antitumor gene in the human bladder.

## 1. Introduction

Transglin/SM22-α/WS3-10/mp27 (TAGLN), a kind of 22-kDa protein, presents primarily in smooth muscle-containing tissues of vertebrates. Cytogenetically, the human TAGLN gene is located to the chromosome 11 q23.2 [1]. When using the TAGLN-deficient mouse embryo model, TAGLN may not be required for the development of the embryo, but plays roles in the morphological transformation of the smooth muscle cell (SMC) [2]. A study has found that decreased levels of TAGLN disrupted normal actin organization leading to the changes in the motile behavior of REF52 fibroblast cells [3]. The depletion of TAGLN resulted in an increase in the capacity of cells to go initiate spontaneous podosome formation, with a concomitant increase in the ability of invasion from Matrigel assays; therefore, TAGLN seemed to be a marker of active stromal remodeling in vicinity of invasive carcinomas [4].

The reduction in the expressions of TAGLN are often found in tumor cell lines, and the TAGLN depletion increases actin dynamics and enhances tumorigenic phenotypes of the cells [3]. Early study indicated that abolition of TAGLN expression is an important early event in tumor progression and a diagnostic marker for breast and colon cancer development [5]. However, issues concerning the tumorigenesis of TAGLN in different tissues are still in controversy. Studies have reported that TAGLN is an antitumor gene in esophageal squamous cell carcinoma and regarded as an oncogene for gastric cancer [6,7,8]. Other studies indicated that TAGLN exerts an anti-metastasis effect in colon and colorectal cancers [9,10,11,12,13]. However, contrary results from different independent laboratories showed that overexpression of TAGLN causes a poor prognosis in colon cancer in vivo and contributes to colorectal cancer progression and metastasis [14,15]. Similar, contrary reports are also found in studies of lung cancer [16,17].

An early study of rabbit bladders suggested that TAGLN is an SMC-lineage marker [18]. However, previous studies indicated that *TAGLN* is one of the common differentially-expressed genes which is significantly decreased in bladder cancer compared with normal bladder tissues [19,20]. The precise functions and the regulatory mechanisms of *TAGLN* in the bladder carcinoma cells are still not illustrated and explored.

In this study, we determined the expressions of *TAGLN* in both bladder carcinoma cells and bladder tissues, and examined the regulatory mechanisms and potential functions of *TAGLN* in bladder carcinoma cells.

## 2. Results

### 2.1. Expressions of TAGLN in Bladder Smooth Muscle Cells, Fibroblast Cells, Normal Epithelial Cells, and Carcinoma Cells

To understand the expression of TAGLN in human bladder cells, we compared levels of TAGLN in human normal primary bladder epithelial cells (HBdEC), bladder smooth muscle cells (HBdSMC), bladder stromal fibroblasts (HBdSF), and four lines of cultured bladder carcinoma cells (RT4, HT1376, T24, and TSGH-8301). Results of RT-qPCR assays revealed that levels of *TAGLN* were higher in both HBdSMC and HBdEC cells than the bladder carcinoma cells (Figure 1A). Further immunoblot assays showed that T24 cells expressed the highest TAGLN protein levels among the four carcinoma cell lines (Figure 1B) which were similar to the results of RT-qPCR assays presented in the Figure 1A. The immunoblot assays also revealed that HBdSMC cells expressed higher protein levels of alpha-smooth muscle actin (α-SMA), and HBdEC cells exhibited higher protein levels of uroplakin-2 (UPK-2), a marker of bladder transitional cells (Figure 1C). The normal primary bladder epithelial cells (HBdEC) presented far higher TAGLN protein levels in comparison to the bladder carcinoma T24 cells (Figure 1D).

### 2.2. Expressions of TAGLN in Paired Human Bladder Tissues

The RT-qPCR analysis of paired human bladder tissues showed that means of ΔΔC_t_ between normal and cancer tissues were 3.77 ± 0.67 using *β-actin* as internal control (Figure 1E) and 4.33 ± 0.72 using *18S* as internal control (Figure 1F), respectively, suggesting significantly higher expressions of *TAGLN* mRNA levels in normal bladder tissues than that in bladder cancer tissues.

### 2.3. TAGLN’s Localization is Predominantly Cytosolic and with F-actin

In order to understand the subcellular location of TAGLN in the bladder carcinoma cells, we transiently overexpressed TAGLN in HT1376 cells. Results of immunofluorescence staining indicated that HT-TAGLN cells expressed higher protein levels of TAGLN, located predominantly in the cytosol, in comparison to HT-DNA cells (Figure 2A,E). The cells were also stained with Texas Red X-Phalloidin to determine the F-actin (Figure 2B,F), and DAPI to highlight the nuclei of HT-DNA and HT-TAGLN cells (Figure 2C,G). Results of immunofluorescence indicated that TAGLN expression colocalized with F-actin (Figure 2D,H). Further study of immunoblot assays with subcellular extraction confirmed the ectopic-TAGLN expression in HT1376 cells (HT-TAGLN). Expression was predominantly present in the cytoplasm, with a little expression in the membrane. TAGLN did not express in the nuclei of HT-DNA and HT-TAGLN cells (Figure 2I).

### 2.4. The Effects of TAGLN on Cell Proliferation in Bladder Carcinoma Cells

Using immunoblot and RT-qPCR assays, we confirmed that expression of *TAGLN* was about 50% in the TAGLN-knockdown T24 cells compared to mock-knockdown (T24_shCOL) cells (Figure 3A). The results of EdU flow cytometry (Figure 3C) and EdU staining proliferation (Figure 3D) assays showed that knockdown of TAGLN (T24_shTAGLN) increased 10% and 9% of cells with EdU-staining compared to mock-transduced (T24_shCOL) cells, respectively. Opposite results were found for ectopic overexpression of TAGLN in HT1376 cells. Figure 3B confirmed the ectopic overexpression of TAGLN in HT1376 (HT-TAGLN) cells compared to mock-transfected HT1376 (HT-DNA) cells, as determined by immunoblot and RT-qPCR assays. When cells were overexpressed the TAGLN, the percentage of positive cells with EdU incorporation were decreased, which was determined by flow cytometry (Figure 3E) and EdU staining proliferation (Figure 3F) assays. Decreasing cell proliferation after ectopic overexpression of TAGLN was also found in the TAGLN-overexpressed HT1376 and TSGH-8301 cells, respectively, determined by Ki67 proliferation assays (Appendix A).

### 2.5. The Effects of TAGLN on Cell Invasion in Bladder Carcinoma Cells

The results of Matrigel invasion assays indicated that the knockdown of TAGLN resulted in a two-fold increase in invasion capacity compared to T24_shCOL cells (Figure 4A). On the contrary, the invasion capacity was downregulated 45% in HT-TAGLN cell compared to HT-DNA cells (Figure 4B). We further cloned the TAGLN-overexpressed TSGH-8301 (8301-TAGLN) cells (Figure 4C) and the TAGLN-knockdown TSGH-8301 (8301_shTAGLN) cells (Figure 4D) to confirm the expression of TAGLN by immunoblot and RT-qPCR assays. The Matrigel invasion assays showed that 8301-TAGLN cells expressed markedly lower invasive capacity than 8301-DNA cells (Figure 4E), while the knockdown of TAGLN in bladder carcinoma TSGH-8301 cells enhanced cell invasion (Figure 4F). Results of wound healing assays also revealed the decreased migration ability of HT1376 cell after the ectopic overexpression of TAGLN (Appendix A).

### 2.6. The Effect of the Ectopic Overexpression of TAGLN on the Tumorigenesis of Bladder Carcinoma HT1376 Cells

The effect of TAGLN on tumor growth in vivo was evaluated by using xenografts in BALB/cAnN-Foxn1^NU^ mice. Tumors generated from HT-DNA cells (Figure 5A) grew faster than those derived from HT-TAGLN cells (Figure 5B). There were no significant differences in the mean body weights of animals between the two groups (Figure 5C). Additionally, tumors generated from HT-TAGLN cells were approximately 26% smaller than the tumors generated from HT-DNA (93.11 ± 19.73 versus 364.13 ± 60.83 mm^3^) after 46 days of growth (Figure 5D). The weight of tumors derived from HT-DNA cells was about four times that from the group of HT-TAGLN cells (0.40 ± 0.07 versus 0.11 ± 0.02; Figure 5E). We randomly selected five tissues from each group to perform RT-qPCR assays, and the results confirmed that TAGLN was overexpressed in the xenograft tumors derived from HT-TAGLN cells (Figure 5F).

### 2.7. p53 and PTEN Upregulated TAGLN Expression in Bladder Carcinoma Cells

The immunoblot (Figure 6A) and RT-qPCR (Figure 6B) assays confirmed that the overexpression of p53 induced TAGLN expression in HT1376 cells. Further, the immunoblot assays showed that camptothecin, a topoisomerase inhibitor, induced not only p53 but also TAGLN expressions, while p53-knockdown attenuated these effects in p53-wild type RT-4 cells (Figure 6C). Similar results found in the reporter assays indicated that a transiently cotransfected-p53 expression vector induced the reporter activity of *TAGLN* reporter vector containing the 5′-flanking region of the human *TAGLN* gene in HT1376 cells (Figure 6D). Further immunoblot assays revealed that the ectopic overexpression of PTEN in T24 cells blocked AKT phosphorylation but enhanced TAGLN expression, whereas PTEN knockdown in RT-4 cells reversed those effects (Figure 6E). Results of RT-qPCR assays showed that ectopic overexpression of PTEN induced *TAGLN* gene expression in T24 cells (Figure 6F), while PTEN-knockdown downregulated *TAGLN* expression in RT-4 cells (Figure 6G). Further reporter assays revealed that a transiently cotransfected-PTEN expression vector upregulated the reporter activity of human *TAGLN* reporter vector in HT1376 cells (Figure 6H). Collectively, TAGLN is upregulated by p53 and PTEN in bladder carcinoma cells.

## 3. Discussion

Bladder cancer is the ninth most common cancer and ranks 13th in terms of deaths worldwide, with the highest incidence rates observed in men around the world [21]. In the United States, bladder cancer is the sixth most common cancer with an estimated 79,030 new cases and 16,870 deaths in 2017 [22]. It seems critical nowadays to explore a new therapeutic molecular target for patients with bladder cancer, since multiple factors are associated with bladder cancer [23,24].

TAGLN may play roles in the morphologic transform process in smooth muscle cells, although TAGLN is not required for the normal development of mouse embryos [2]. Overexpression of TAGLN was strictly limited to the regulation of the tumor-induced, reactive, myofibroblastic stromal tissue, compartment-specific cell type’s expression in tumoral stroma compared to neoplastic epithelial cells [4]. The TAGLN depletion increased the capacity of cells to form podosomes spontaneously with a concomitant increase in the ability to invade Matrigel, suggesting that TAGLN may relate to the tumorigenic properties of cells [3]. A prior study indicated that an early loss of *TAGLN* gene expression is important for tumor progression and viewed that as a diagnostic marker for breast and colon cancer development [5]. However, results from the functional assays of TAGLN in different cancers are still inconclusive. Although studies in bladder cancer suggested that *TAGLN* was one of common differentially-expressed genes, significantly decreased in cancer compared with normal tissues [19,20], the precise functions and regulatory mechanisms of *TAGLN* in bladder carcinoma cells are still unexplored.

This study illustrated that protein and mRNA levels of *TAGLN* are higher in human normal primary bladder epithelial cells (HBdEC) than in bladder carcinoma cells. Meanwhile, a recent study showed that *TAGLN* was expressed in bladder carcinoma T24 and SW780 cells at the transcription and translation levels [25]. Although our study suggested that TAGLN expressions among the bladder carcinoma cell lines could be dependent on the cell type but not relevant to the extent of neoplasia in vitro, the RT-qPCR analysis of paired human bladder tissues further demonstrated a significantly lower expression of *TAGLN* mRNA levels in bladder tumors than adjacent normal bladder tissues (Figure 1). These results are in agreement to other studies of badder cancer in vivo [19,20,26], suggesting that *TAGLN* is a tumor suppressor gene in the bladder. The family of transgelins consists of three homologs (TAGLN, transgelin-2, and transgelin-3) in human tissues. A prior study has suggested that TAGLN and trangelin-2 act as cancer biomarkers and are differentially expressed in the tumor and stroma cells [27]. Further studies implied that transgelin-2 has a potentially oncogenic function in bladder cancer, both in vitro and in vivo [28,29]. Our study presents the tumor suppressor characteristics of TAGLN in bladder cancer, both in vitro and in vivo, and agrees with the concept of that the role of transgelin-2 in tumor development might be contradictory to the role of TAGLN [27].

This study also found high levels of TAGLN expressed in the bladder’s smooth muscle cells (HBdSMC) and stromal fibroblasts (HBdSF) in vitro (Figure 1). Earlier study indicated that TAGLN is an SMC-lineage marker in rabbit bladders [18]; however, TAGLN was regarded as a marker of active stromal remodeling in the vicinity of invasive carcinomas as well [4]. In the prostate study, TAGLN levels were found to be elevated in the stroma but decreased in the carcinomic epithelial cells during cancer progression [30,31]. However, further investigation into the function of TAGLN in the stroma of bladders is still necessary.

Our finding indicated that the expression of TAGLN was predominant in the cytosol and associated with F-actin during the ectopic overexpression of bladder carcinoma HT1376 cells. Further immunoblot assays with subcellular extraction confirmed that TAGLN did not express in the nuclei but in the cytoplasm, mostly with a little expression in the membrane (Figure 2). These results are in agreement with previous studies, which implied that TAGLN is localized to the cytosol where it binds to F-actin [32].

Early studies suggested that *TAGLN* was a tumor suppressor gene in various cancers, while some cancers possessed the oncogenic characteristic of *TAGLN*. The overexpression of TAGLN potentially contributed to the progression and metastasis of colorectal cancer; meanwhile, high levels of TAGLN related to a poor prognosis of colon cancer in vivo [14,15]. TAGLN was upregulated in cell lines from the human lung adenocarcinoma under hypoxic conditions that caused the migration ability of the tumor cells; moreover, a high TAGLN expression correlated with an advanced TNM stage, lymph node metastasis, and greater differentiation in lung adenocarcinoma [16]. However, other studies concluded that *TAGLN* is a tumor suppressor gene associated with a poor prognosis in colorectal carcinoma patients [9,10]. An in vitro study declared that apigenin upregulated the expression of TAGLN in mitochondria to exert its anti-tumor growth and anti-metastasis effects in colorectal cancer [11]. The restoration of the TAGLN-induced inhibition of colon carcinogenesis in vivo and in vitro suggested that TAGLN might potentially function as a novel tumor suppressor [12,13]. A previous study of prostates also identified *TAGLN* as a tumor suppressor gene in vivo and in vitro [31], and others found the same in bladder cancer, based on a gene profile analysis [19,20,26]. Our study is the first study providing direct evidence to demonstrate that *TAGLN* is an antitumor gene in the bladder carcinoma cells. Results from this study revealed that the ectopic overexpression of TAGLN decreased cell proliferation, invasion, and migration in bladder carcinoma cells in vitro, while the knockdown of *TAGLN* reversed the effects (Figure 3, Figure 4, Appendix A). The anti-tumorigenesis characteristics of *TAGLN* in bladder carcinoma cells was also demonstrated by an in vivo xenograft animal study (Figure 5).

Previous studies have indicated that *TAGLN* is a TGFβ-inducible gene of human skeletal stem cells, prostate carcinoma cells, and prostate fibroblast cells [31,33,34]. Steroid receptor coactivatior and mir-145 also modulated *TAGLN* expression in different cells [35,36]. Interestingly, *TAGLN* was regarded as an ARA54-associated AR inhibitor that suppressed AR function, and induced the apoptosis of human prostate LNCaP cells through its interaction with p53 [37]. Our study is the first to reveal that *TAGLN* is the downstream gene of *p53* and *PTEN* in bladder carcinoma cells. *p53* is a well-known tumor suppressor gene and *p53* mutations correlate with a variety of human cancers [38]. The overexpression of p53 in p53-mutant HT1376 cells, or with camptothecin (a topoisomerase I inhibitor) treatment in p53-wild type RT-4 cells, induced TAGLN expression, while p53-knockdown attenuated this effect. It was found that camptothecin induced p53 expression in RT-4 cells in a previous study that used bladder carcinoma cells [39]. Two putative p53 response elements (5′-AGGCAAGTTCTGTGTTAGTCATGCAC-3′, –2151 to –2126; and 5′-AAACTTGTTTTTATAGCTCTG CTTGAAG-3′, –1956 to –1929), which are similar to the p53 response element consensus sequence (RRRCWWGYYYN_0–13_RRRCWWGYYY), as previously reported [40], were found in the promoter region of the human *TAGLN* gene—determined by in silico analysis. The precise mechanisms of the modulation *TAGLN* gene expression by p53 still need to be further investigated.

Phosphatase and tensin homolog deleted on chromosome 10 (*PTEN*) has been widely known as a tumor suppressor gene, and *PTEN* mutation or deletion is frequently noted in a lot of cancers, including bladder [39,41,42]. The most known function of PTEN is as the negative regulator of the PI3K/Akt/mTOR pathway, which is a crucial signal transduction pathway for cancer cell growth [39,43]. For bladder cancer, a loss of PTEN expression has been correlated with the cell growth and the invasion of bladder carcinoma [39,41]. In this study, we found that ectopic overexpression of PTEN in PTEN-mutant T24 cells blocked Akt phosphorylation but induced *TAGLN* expression, while knockdown of *PTEN* in PTEN-wildtype RT-4 cells reversed those effect (Figure 6). Conclusively, our study confirmed that *TAGLN* is a target gene of p53 and PTEN, and should be referred to as a tumor suppressor in bladder carcinoma cells, both in vitro and in vivo.

## 4. Materials and Methods

### 4.1. Cell Cultures and Chemicals

The four bladder-transitional carcinoma cell lines, RT-4, HT1376, TSGH-8301, and T24 cells, were purchased from the Bioresource Collection and Research Center (BCRC, Hsinchu, Taiwan) as described previously [44]. Human bladder smooth muscle cells (HBdSMC), bladder stromal fibroblasts (HBdSF), smooth muscle cell medium, and fibroblast medium were purchased from ScienCell Research Laboratories Inc. (Carlsbad, CA, USA). Human normal primary bladder epithelial cells (HBdEC; ATCC PCS-420-010) were purchased from American Type Culture Collection (ATCC; Manassas, VA, USA) and maintained in the prostate epithelial cells’ basal mediums (ATCC PCS 440-030). DAPI (4,6-diamino-2-phenylindole) and camptothecin were obtained from Sigma-Aldrich Co. (St. Louis, MO, USA). Fetal calf serum (FCS) was from HyClone (Logan, UT, USA), and RPMI 1640 media was obtained from Invitrogen (Carlsbad, CA, USA).

### 4.2. Tissue Collection and Analysis

The specimens of the human paired bladder tissue biopsies were obtained from patients admitted to the Department of Urology, Chang Gung Memorial Hospital-Linkou (Tao-Yuan, Taiwan). Bladder tissues were classified based on the pathological examinations of the parallel preparations from respective samples by attending pathologists. The Institutional Review Board of the Chang Gung Memorial Hospital approved the protocol for tissue collection and analysis (IRB 201800981B0, 1 September 2018).

### 4.3. Expression Vector and Stable Transfection

The full-length human *TAGLN* cDNA (HG14991-UT) was purchased from Sino Biologic Inc. (Beijing, China). Electroporation was conducted using an ECM 830 Square Wave Electroporation System (BTX, San Diego, CA, USA), set at 180 V (for TSGH-8301 cells) or 190 V (for HT1376 cells), with a 70-msec pulse length and one pulse setting. Transfected cells (8301-TAGLN and HT-TAGLN) were selected by 100 μg/mL of hygromycin (Sigma-Aldrich Co.). For construction of the mock-transfected cells (8301-DNA and HT1376-DNA), cells were transfected with a controlled pCMV3 expression vector (Sino Biologic Inc.) and were clonally selected in the same manner as described above. The *p53* and *PTEN* expression vectors were constructed and p53-overexpressed HT1376 (HT-p53) cells and PTEN-overexpression T24 (T24-PTEN) cells were cloned as described previously [39].

### 4.4. Knock-Down TAGLN, p53, and PTEN

TSGH-8301 and T24 cells were transduced with control shRNA lentiviral particles-A (Sc-108080) or transgelin shRNA (h) Lentiviral Particles (sc-44163-V) that were purchased from Santa Cruz Biotechnology, Inc. (Santa Cruz, CA, USA). Two days after transduction, the cells (8301_shCOL, 8301_shTAGLN, T24_shCOL, and T24_shTAGLN) were selected by incubation with 10 μg/mL puromycin dihydrochloride for at least another 5 generations. The p53-knockdown RT-4 (RT4_shp53), PTEN-knockdown RT-4 (RT4_shPTEN), and mock-knockdown RT-4 (RT4_shCOL) cells were cloned as previously described [39].

### 4.5. Immunoblot Assays

Cells were cultured in RPMI-1640 medium with 10% FCS for 48 h. The nuclear and cytoplasmic fractions were separated using the subcellular protein fractionation kit (Thermo Fisher Scientific Inc., Rockford, NJ, USA) as described previously [44]. Equal amounts of whole cell, membrane, nuclear, or cytoplasmic lysates were separated on a 10%–12% SDS-polyacrylamide gel and assayed by the Western lightning plus-ECL detection system, as instructed by the manufacturer (PerkinElmer Inc, Waltham, MA, USA). The blotting membranes were probed with the antisera of p53 (DO-1, Santa Cruz Biotechnology), PTEN (#9552, Cell Signaling Technology, Danvers, MA, USA), Akt (#4691, Cell Signaling Technology), phopsho-Akt^S473^ (#9271, Cell Signaling Technology), or β-actin antiserum (MAB1501, Merck Millipore, Burlington, MA, USA). The intensities of different bands were recorded using the LuminoGraph II (Atto corporation, Tokyo, Japan).

### 4.6. Real-Time Reverse Transcriptase-Polymerase Chain Reaction (RT-qPCR)

Total RNA from tissues or cells was isolated using Trizol reagent (Ambion, Life Technologies, Carlsbad, CA, USA). The cDNA was synthesized using Superscript III pre-amplification system (Invitrogen), as described previously [45]. The PCR probes for human *p53* (Hs01034249_m1), *PTEN* (Hs02621230_Sl), *TAGLN* (Hs01038777_g1), *β-actin* (Hs01060665_g1), and *18S* (Hs03003631_g1) were purchased from Applied Biosystems (Foster City, CA, USA). Real-time polymerase chain reactions (qPCR) were performed using an CFX Connect Real-Time PCR system (Bio-Rad Laboratories, Foster city, CA, USA) and the mean cycle threshold (C*_t_*) values were calculated for internal control and target genes, as described previously [44].

### 4.7. Cell Immunofluorescence and F-Actin Staining

Cells were seeded onto the glass bottom culture dishes (MatTek, Ashland, MD, USA) which were precoated with 50 μL fibronectin. Cells were fixed at room temperature in 4% paraformaldehyde in phosphate buffered saline (PBS) for 20 min after 24 h of attachment. After that, cells were permeabilized for 10 min in a 0.2% solution of Triton X-100 and blocked in 1% BSA for 1 h after being washed with PBS. The coverslips were incubated with 1:100 diluted anti-TAGLN antiserum overnight at 4 °C and then incubated with donkey anti-rabbit secondary antibody (A21206, Invitrogen) for 1 h. The F-actin protein expression was revealed by incubation with Texas Red X-Phalloidin (Invitrogen). The coverslips were mounted with the Prolong Gold antifade reagent with DAPI (Thermo Fisher Scientific Inc.). The immunofluorescence was examined using a confocal microscope (LSM510 Meta, Zeiss, Oberkochen, Germany) as described previously [46].

### 4.8. EdU Staining Proliferation Assay

Cells were seeded onto the sterile glass coverslips. After 48 h of attachment, EdU was added to the culture medium to a final concentration of 50 μM for another 2 h. Cells were washed and fixed in 3.7% paraformaldehyde in phosphate buffered saline (PBS) at room temperature for 15 min, and then washed twice with 200 μL of 3% BSA in PBS. Cells were permeabilized with saponin-based permeabilization and washing reagent for 20 min, and then incubated for 30 min with Click-iT reaction cocktail, containing fluorescent dye azide, CuSO_4_, PBS, and reaction buffer (Thermo Fisher Scientific Inc.). Cells were washed twice with 200 μL of 3% BSA in PBS, and the coverslip was mounted with Prolong Gold antifade reagent with DAPI. The EdU fluorescence was recorded and photographed under a microscope (BX43, OLYMPUS, Tokyo, Japan).

### 4.9. EdU Flow Cytometry Assay

Cells (5 × 10^5^) were serum starved for 24 h, and then incubated with RPMI1640 medium with 10% serum for another 48 h. EdU (5-ethynyl-2′-deoxyuridine; 10 μM) was added to the culture medium for further 2 h. Cells were collected, fixed, and permeabilized with saponin-based permeabilization and washing reagent for 20 min, and then incubated for 30 min with Click-iT reaction cocktail (Thermo Fisher Scientific Inc.), as described by the manufacturer. The EdU fluorescence of cells was detected using Attune NxT acoustic focusing cytometer (Thermo Fisher Scientific Inc.).

### 4.10. Ki67 Proliferation Assay

Cells were serum-starvated for 24 h, and then incubated with RPMI1640 medium with 10% serum for further 48 h. Cells were stained with PE Mouse Anti-Ki-67 kit (catalogue number 556027; BD Biosciences, Bedford, MA, USA) as described by the manufacturer. Briefly, cells were washed, pelleted, and fixed with cold, 70% ethanol at −20 °C overnight. The PE-conjugated anti-Ki-67 antibody and PE-conjugated mouse IgG1 were added to the cells and the mixture was incubated at room temperature for further 30 min in the dark, after the cells had been washed twice with washing buffer (PBS with 1% FBS, 0.09% NaN_3_). The cells were washed with washing buffer and re-suspended in PBS, and then the Ki67 fluorescence of cells were detected using Attune NxT acoustic focusing cytometer.

### 4.11. Cell Migration Assay

The cell migration analysis was determined by a wound healing assay, as described previously [46]. In brief, the cell sheets were wounded with a plastic pipette tip when they attained a confluent monolayer. The wound closure (the gap width) was photographed microscopically (IX71, Olympus, Tokyo, Japan) with a digital camera during the indicated times. The quantification of cell migration was determined within a defined area using the Image J program.

### 4.12. Matrigel Invasion Assay

The in vitro invasion ability of cells was determined by the Matrigel invasion assay (BD Biosciences). Cells that invaded into other side of the membrane were fixed using 4% paraformaldehyde and then stained with 0.1% crystal violet solution for 10 min. The quantity of cells that invaded the Matrigel was recorded microscopically (IX71, Olympus, Tokyo, Japan) and the images were analyzed as described previously [44].

### 4.13. Xenograft Animal Model

The performance of animal studies was approved by the Chang Gung University Animal Research Committee (Approval No.: CGU107-092, 1 September 2018). The male nude mice (BALB/cAnN-Foxn1) were obtained from the animal center of National Science Council, Taiwan, at 4 weeks old. In all the procedures, every effort was made to minimize the suffering of the laboratory animals in accordance with the United States’ National Institutes of Health Guide for the Care and Use of Laboratory Animals. The cells were detached with Gibco^Tm^ Versene solution and washed with RPMI 1640 medium with 10% FCS, and then were re-suspended in a PBS solution. The mice were anesthetized intraperitoneally and 1 × 10^6^ cell/100 μL cells were injected subcutaneously on the right or left lateral back wall in close proximity to the shoulder of each mouse. The growth of the xenografts was measured by vernier caliper measurements on the days indicated. The tumor volume was determined by the formula volume = π/6 × W × D^2^, as described previously [47].

### 4.14. TAGLN Reporter Vector and Reporter Assays

The 5′-DNA fragment (−1 to −4573) of the human *TAGLN* gene, according to the sequence from GenBank (AP005018.1), was synthesized by Invitrogen. The human *TAGLN* reporter vector was constructed by cloning the DNA fragment into the pbGL3 reporter vector (Promega Biosciences, Madison, WI, USA) with *Hind III* sites. Proper ligation was confirmed by extensive restriction mapping and sequencing. The cells were plated onto 24-well plates at 1 × 10^4^ cells/well for 1 day prior to transfection. Cells were transiently cotransfected with reporter vectors and expression vectors as indicated, using the X-tremeGene HP DNA transfection reagent (Roche Diagnostics GmbH, Mannheim, Germany), as described previously [48]. The luciferase activity was adjusted for transfection efficiency using the normalization control plasmid pCMVSPORTβgal.

### 4.15. Statistical Analysis

All the results are expressed as the means ± S.Es. Statistical significance was determined by one-way ANOVA and Student’s *t* tests using the SigmaPlot 10.0 (SPSS Inc., Chicago, IL, USA). The * represented the *p* < 0.05 and the ** represented the *p* < 0.01.

## 5. Conclusions

Our results indicated that the expression of transgelin, a regulator of the actin cytoskeleton, is higher in bladder normal epithelial cells than in carcinoma cells. Our experiments provided evidence suggesting that *TAGLN* is a p53 and PTEN-downstream gene, which attenuates cell proliferation and invasion in vitro, and tumorigenesis in vivo. *TAGLN* seems to function as a tumor suppressor gene in bladder carcinoma cells.

## Figures and Tables

**Figure 1 ijms-20-04946-f001:**
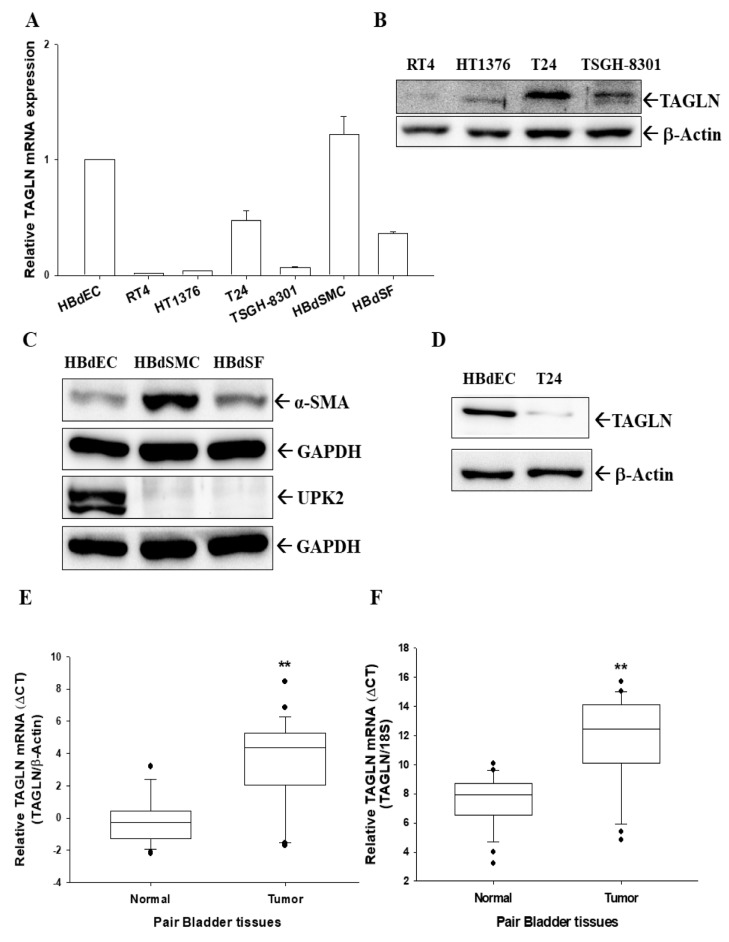
Gene expression of *TAGLN* in human bladder cells and tissues. (**A**) Total RNA from bladder smooth muscle cells (HBdSMC), fibroblast cells (HBdSF), normal epithelial cells (HBdEC), and carcinoma cell lines (RT4, HT1376, T24, and TSGH-8301) were extracted for RT-qPCR (± SE; *n* = 3) assays. (**B**) Bladder carcinoma cell lines (RT4, HT1376, T24, and TSGH-8301) were lysed, and TAGLN and β-actin were determined by immunoblotting. (**C**) HBdEC, HBdSMC, and HBdSF cells were lysed and α-SMA, UPK2, and GAPDH were determined by immunoblotting. (**D**) HBdEC and T24 cells were lysed and TAGLN and β-actin were determined by immunoblotting. Quantitative analysis of TAGLN expression in paired bladder cancerous and normal tissues was determined by RT-qPCR using the β-actin (**E**) or 18S (**F**) as the internal control. Box plot analysis was used to compare the TAGLN expressions in cancerous and normal bladder tissues (*n* = 25). ** represented the *p* < 0.01.

**Figure 2 ijms-20-04946-f002:**
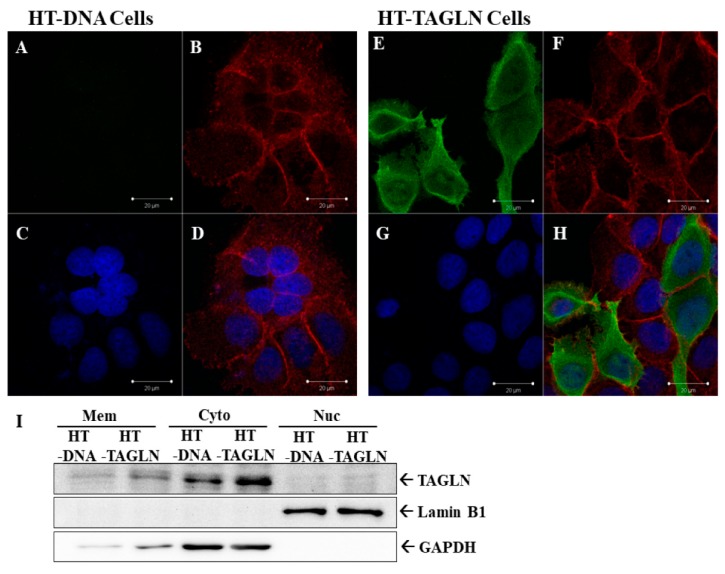
Immunofluorescence staining of TAGLN in TAGLN-overexpressed-HT1376 cells. Immunofluorescence detection of TAGLN in mock-transfected HT1376 (HT-DNA) (**A**) and TAGLN-overexpressed HT1376 (HT-TAGLN) (**E**) cells. The red-stained with Texas Red X-Phalloidin represents the F-actin (**B**,**F**). The blue-stained by DAPI represents nuclei of cells (**C**,**G**). All images were observed and recorded under the same settings with a confocal microscope. Immunofluorescence results indicated that TAGLN expression colocalized with F-actin (**D**,**H**). (**I**) Cells were subcellularly extracted into the membrane (Mem), cytosol (Cyto), and nuclei (Nuc). The protein levels of TAGLN, Lamin B, and GAPDH were determined using immunoblot assays.

**Figure 3 ijms-20-04946-f003:**
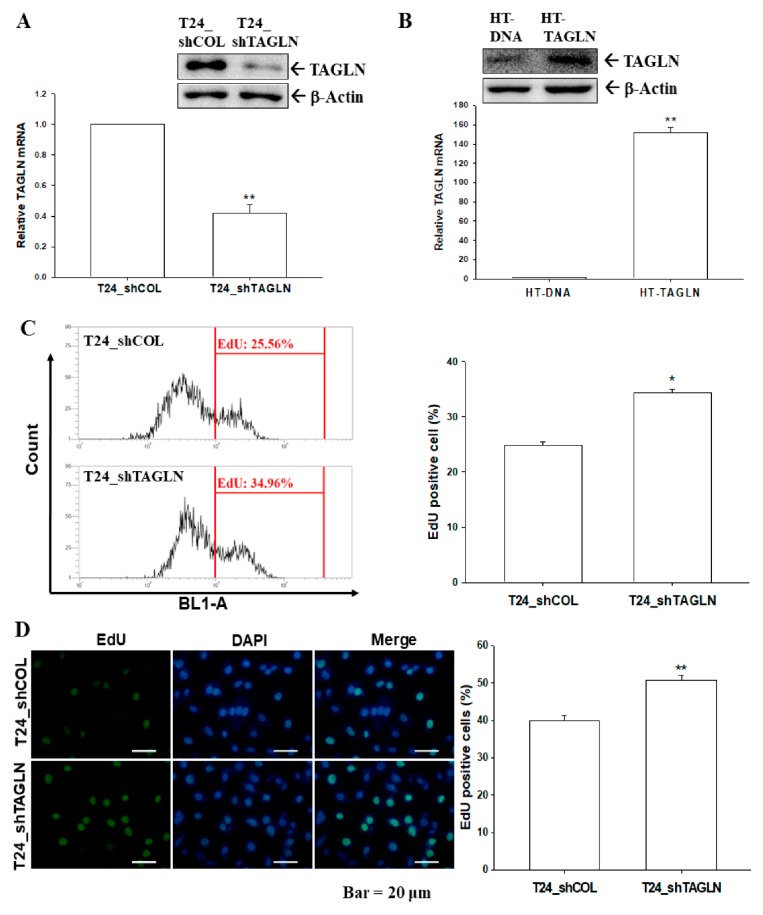
The effects of TAGLN on the cell proliferation of bladder carcinoma cells. The expressions of TAGLN in T24-knockdown T24 (T24_shTAGLN) cells (**A**) and ectopic TAGLN-overexpressed HT1376 (HT-TAGLN) cells (**B**) in comparison to mock-transfected control cells (T24_shCOL and HT-DNA) were determined by immunoblot (top) and RT-qPCR (bottom) assays (± SE; *n* = 3). The proliferation abilities of T24_shCOL and T24_shTAGLN cells were determined by flow cytometry with Click-iT EdU flow cytometry assays (**C**) and EdU cell immunofluorescence staining (± SE; *n* = 4) (**D**). The proliferation abilities of HT-DNA and HT-TAGLN cells were determined by flow cytometry with (**E**) Click-iT EdU flow cytometry assays and (**F**) EdU cell immunofluorescence staining (± SE; *n* = 4). * represented the *p* < 0.05 and the ** represented the *p* < 0.01.

**Figure 4 ijms-20-04946-f004:**
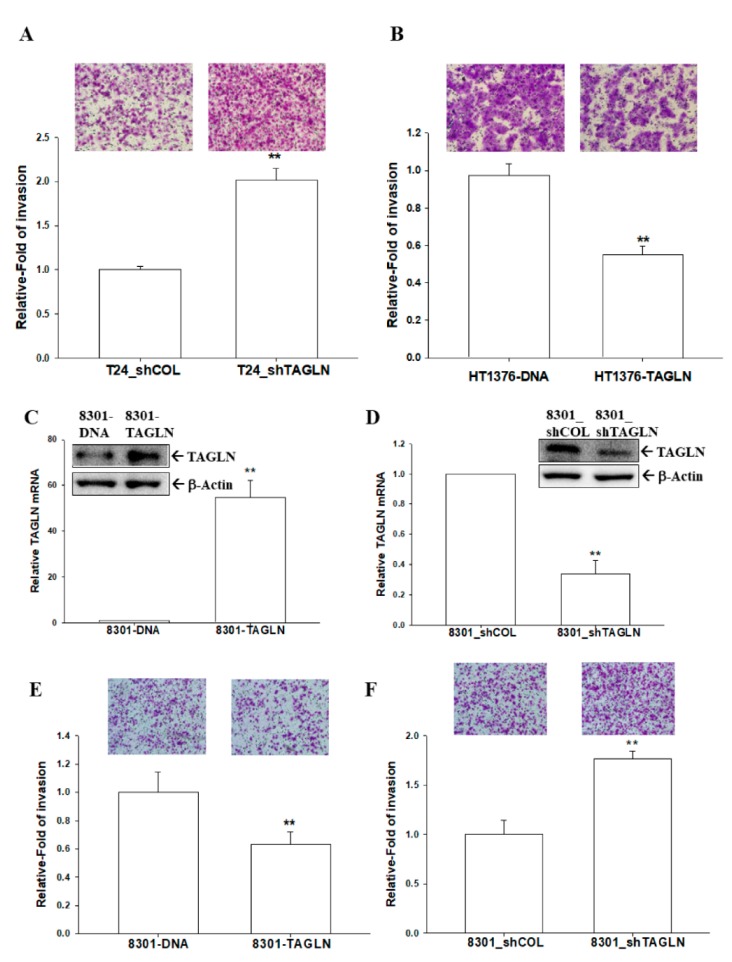
Modulation of TAGLN on cell invasion in bladder carcinoma cells. The invasion abilities of cells were determined by in vitro Matrigel invasion assays. Data are presented as mean percentages (± SE; *n* = 3) in relation to those of the T24_shCOL (**A**) or HT-DNA (**B**) groups. (**C**) Expressions of TAGLN in mock-transfected TSGH-8301 (8301-DNA) and ectopic TAGLN overexpression TSGH-8301 (8301-TAGLN) cells were determined by immunoblotting (top) and RT-qPCR (bottom) assays (± SE; *n* = 3). (**D**) Expressions of TAGLN in mock-knockdown TSGH-8301 (8301_shCOL) and TAGLN-knockdown TSGH-8301 (8301_shTAGLN) cells were determined using immunoblot (top) and RT-qPCR (bottom) assays (± SE; *n* = 3). The invasion abilities of cells were determined by in vitro Matrigel invasion assays. Data are presented as mean percentages (± SE; *n* = 3) in relation to those of the 8301-DNA (**E**) or (**F**) 8301_shCOL groups. ** represented the *p* < 0.01.

**Figure 5 ijms-20-04946-f005:**
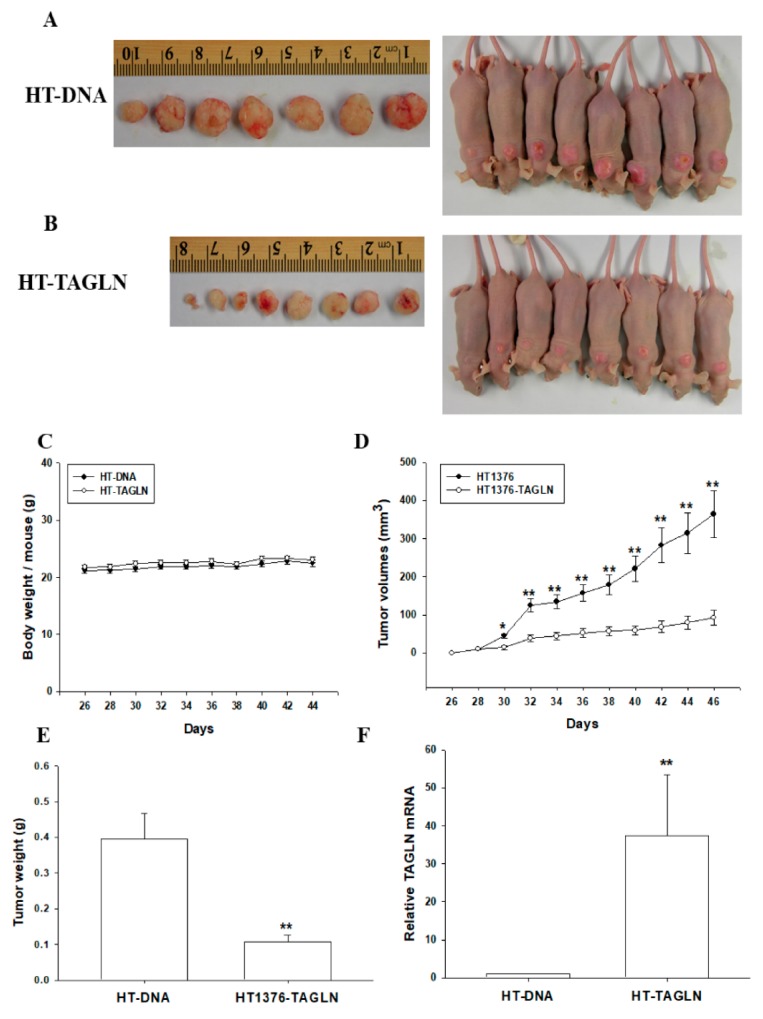
The ectopic overexpression of TAGLN attenuates tumorigenesis, in vivo, for bladder carcinoma HT1376 cells. Four-week-old male athymic nude (nu/nu) mice were randomized into two groups: HT-DNA (*n* = 7; **A**) and HT-TAGLN (*n* = 7; **B**). HT-DNA cells and HT-TAGLN cells (1 × 10^6^) were injected subcutaneously in the dorsal area of the mice, respectively. The animal body weights (**C**) and tumor growth rates (**D**) were measured every 2 days, starting at the 3-weeks-of-growth point (day 26), at which the tumors became perceptible under the skin after inoculation. The tumor weights (**E**) and levels of TAGLN mRNA (**F**) were determined after the animals were sacrificed (± SE; *n* =7). * represented the *p* < 0.05 and the ** represented the *p* < 0.01.

**Figure 6 ijms-20-04946-f006:**
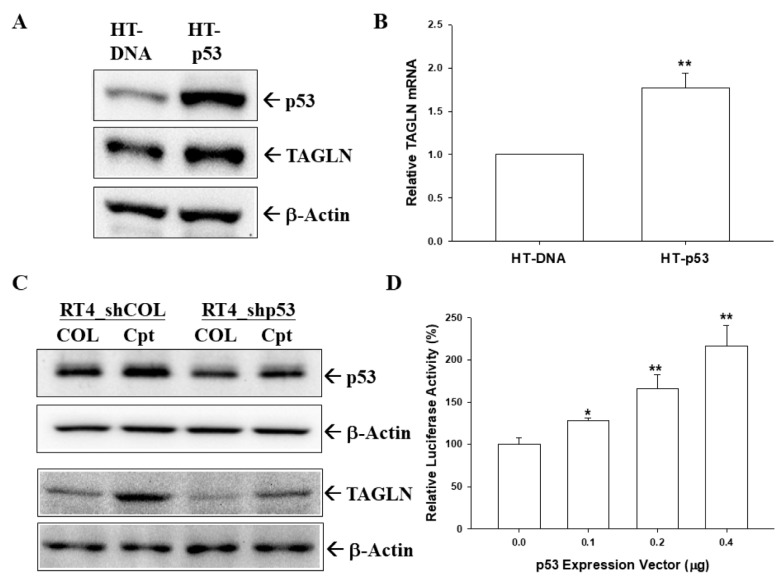
Modulation of p53 and PTEN on TAGLN expression in bladder carcinoma cells. The mock-transfected HT1376 (HT-DNA) and p53-overexpressed HT1376 (HT-p53) cells were lysed, and then TAGLN, p53, and β-actin were determined by immunoblotting (**A**) and RT-qPCR (**B**) assays (± SE; *n* = 3). The mock-knockdown RT4 (RT4_shCOL) and p53-knockdown RT4 (RT4_shp53) cells were treated with camptothecin (1 μM) for 16 h. The cells were lysed, and then TAGLN, p53, and β-actin were determined by immunoblotting (**C**). (**D**) The reporter activity of TAGLN reporter vector cotrasnfected with various dosages of p53, as indicated in HT1376 cells. Data are expressed as the mean percentages ± SEs (*n* = 6) of luciferase activity relative to the mock-transfected group. The mock-transfected T24 (T24-DNA), PTEN-overexpressed T24 (T24-PTEN), mock-knockdown RT4 (RT4_shCOL), and PTEN-knockdown RT4 (RT4_shpPTEN) cells were lysed, and then TAGLN, PTEN, Akt, pAkt^S473^, and β-actin were determined by immunoblotting (**E**). Expressions of PTEN and TAGLN in T24-DNA, T24-PTEN (**F**), RT4_shCOL, and RT4_shPTEN (**G**) cells were determined by RT-qPCR assays (± SE; *n* = 3). (**H**) The reporter activity of TAGLN reporter vector cotrasnfected with various dosages of PTEN, as indicated in HT1376 cells. Data are expressed as the mean percentages ± SEs (*n* = 6) of luciferase activity relative to the mock-transfected group. * represented the *p* < 0.05 and the ** represented the *p* < 0.01.

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
