# Peer review of "Transgelin, a p53 and PTEN-Upregulated Gene, Inhibits the Cell Proliferation and Invasion of Human Bladder Carcinoma Cells In Vitro and In Vivo"

_ijms, 2019, doi:10.3390/ijms20194946_

Round 1

Reviewer 1 Report

The authors have performed and in depth study of TAGLN localization, modulation by chemo drug, p53 and PTEN in bladder cancer. The study is very thorough. The presentation needs major improvement. In particular, the methods and results section. The study is well conducted and detailed, so the manuscript needs to be improved.

where did the authors get primary bladder cells? Fig 6 is really hard to interpret the language in most of the manuscript makes it very challenging to be sure the results being interpreted are what was conducted.Proliferation assay is very difficult to interpret in particular (methods).

Author Response

For reviewer 1: Thanks for your kindly support our manuscript. Grammatical and writing style errors in the original version have been corrected in green font by our colleague who is a native English speaker. We used red font to highlight the modifications in the revised manuscript to indicate where the changes we have made occur in the text. According to the reviewer recommendation, several modifications of this manuscript are listed as following:

The primary normal bladder epithelial cells were obtained from ATCC as we described in the section of 4.1 Cell culture and Chemicals

L311-314: Human normal primary bladder epithelial cells (BdEC; ATCC PCS-420-010) were purchased from American Type Culture Collection (ATCC; Manassas, VA, USA) and maintained in the prostate epithelial cells basal medium (ATCC PCS 440-030).

We agreed reviewer’s suggestion. The description of Figure 6 was reorganized and rewrote in the section of Results 2.7 as following:

L181-194: The immunoblot (Figure 6A) and RT-qPCR (Figure 6B) assays. confirmed that the overexpression of p53 induced TAGLN expression in HT1376 cells.  Further, the immunoblot assays  showed that camptothecin, a topoisomerase inhibitor, induced not only p53 but also TAGLN expressions, while p53-knockdown attenuated these effects in p53-wild type RT-4 cells (Figure 6C). Similar results found in the reporter assays indicated that transiently cotransfected-p53 expression vector induced the reporter activity of TAGLN reporter vector containing the 5’-flanking region of human TAGLN gene in HT1376 cells (Figure 6D). Further immunoblot assays revealed that ectopic overexpression of PTEN in T24 cells blocked AKT phosphorylation but enhanced TAGLN expression; whereas, PTEN knockdown in RT-4 cells reversed these effects (Figure 6E). Results of RT-qPCR assays showed that ectopic overexpression of PTEN induced TAGLN gene expression in T24 cells (Figure 6F), while PTEN-knockdown downregulated TAGLN expression in RT-4 cells (Figure 6G). Further reporter assays revealed that transiently cotransfected-PTEN expression vector upregulated the reporter activity of human TAGLN reporter vector in HT1376 cells (Figure 6H). Collectively, TAGLN is upregulated by p53 and PTEN in bladder carcinoma cells.

We agreed the review’s suggestions and reorganized the description of some methodologies in the section of Materials and Methods as following

L338-340: The p53-knockdowned RT-4 (RT4_shp53), PTEN-knockdowned RT-4 (RT4_shPTEN), and mock-knockdowned RT-4 (RT4_shCOL) cells were cloned as described previously [39]

L342-351: Cells were cultured in RPMI-1640 medium with 10% FCS for 48 h. The nuclear and cytoplasmic fractions were separated using the subcellular protein fractionation kit (Thermo Fisher Scientific Inc., Rockford, NJ, USA) as described previously [44]. Equal amounts of whole cell, membrane, nuclear, or cytoplasmic lysis were separated on a 10-12% SDS-polyacrylamide gel and assayed by the Western lightning plus-ECL detection system described by the manufacturer (PerkinElmer Inc, Waltham, MA, USA). The blotting membranes were probed with antiserum of p53 (DO-1, Santa Cruz Biotechnology), PTEN (#9552, Cell Signaling Technology, Danvers, MA, USA), Akt (#4691, Cell Signaling Technology), phopsho-AktS473 (#9271, Cell Signaling Technology), or β-actin antiserum (MAB1501, Merck Millipore, Burlington, MA, USA). The intensities of different bands were recorded using the LuminoGraph II (Atto corporation, Tokyo, Japen).

L353-360: Total RNA from tissues or cells was isolated using Trizol reagent (ambion, Life Technologies). The cDNA was synthesized using Superscript III pre-amplification system (Invitrogen) as described previously [45]. The PCR probes for human p53 (Hs01034249_m1), PTEN (Hs02621230_Sl), TAGLN (Hs01038777_g1), -actin (Hs01060665_g1), and 18S (Hs03003631_g1) were purchased from Applied Biosystems (Foster City, CA, USA). Real-time polymerase chain reactions (qPCR) were performed using an CFX Connect Real-Time PCR system (Bio-Rad Laboratories, Foster city, CA, USA) and the mean cycle threshold (Ct) values were calculated for internal control and target genes as described previously [44].

L361-371: 4.7. Cell Immunofluorescence and F-actin Staining

Cells were seeded onto the glass bottom culture dishes (MatTek, Ashland, MD, USA) which precoated with 50 μl fibronectin. Cells were fixed at room temperature in 4% paraformaldehyde in phosphate buffered saline (PBS) for 20 min after 24 h of attachment. After that, cells were permeabilized for 10 min in a 0.2% solution of Triton X-100 and blocked in 1% BSA for 1 h after washed with PBS. The coverslips were incubated with 1:100 diluted anti-TAGLN antiserum overnight at 4˚C and then incubated with donkey anti-rabbit secondary antibody (A21206, Invitrogen) for 1 h. The F-actin protein expression was revealed by incubation with Texas Red X-Phalloidin (Invitrogen). The coverslips were mounted with Prolong Gold antifade reagent with DAPI (Thermo Fisher Scientific Inc.). The immunofluorescence was examined using confocal microscope (LSM510 Meta, Zeiss, Oberkochen, Germany) as described previously [46].

L383-389: Cells (5 x 105) were serum starved for 24 h, and then incubated with RPMI1640 medium with 10% serum for another 48 h. EdU (5-ethynyl-2’-deoxyuridine; 10 mM) was added to the culture medium for further 2 h. Cells were collected, fixed, and permeabilized with saponin-based permeabilization and washing reagent for 20 min, and then incubated for 30 min with Click-iT reaction cocktail (Thermo Fisher Scientific Inc.) as described by the manufacturer. The EdU fluorescence of cells was detected using Attune NxT acoustic focusing cytometer (Thermo Fisher Scientific Inc.).

L402-403: The wound closure (the gap width) was photographed microscopically (IX71, Olympus, Tokyo, Japan) with a digital camera during the indicated times.

L416-418: . The cells were detached with GibcoTm Versene solution and washed with RPMI 1640 medium with 10% FCS, and then were re-suspended in a PBS solution.

Reviewer 2 Report

The manuscript studied expression pattern of TAGLN in bladder cancers and its roles in selected cancer cells. Study also suggest that TAGLN is p51 and PTEN-regulated genes. TAGLN expressions were higher in bladder smooth muscle cells, fibroblast cells, and normal epithelial cells than carcinoma cells. RT-qPCR revealed that TAGLN expressions were higher in normal tissues than the paired tumor tissues. TAGLN knockdown enhanced cell proliferation and invasion while overexpression of TAGLN had the inverse effects in several bladder carcinoma cells in vitro and in vivo. Ectopic overexpression of either p53 or PTEN induced TAGLN expression, while p53 knockdown downregulated TAGLN expression in bladder carcinoma. This manuscript is interesting to show that TAGLN is bladder cancer cell growth inhibitor and might be associated with infamous tumor suppressors such as p53 and PTEN. Major drawback for this manuscript to this reviewer is that data presentation is not clear and not consistent with poor figure quality. This article also needs to provide detailed information on TAGLN relevance to p53 and PTEN. Or authors conclusion that “TAGLN is a p53- and PTEN-upregulated gene” is premature. I am providing details below. Major issues: 1. If authors want to claim that TAGLN is regulated by either p53 or PTEN, they needs to show that functional or physical interaction between TAGLN and those signaling pathways, not just expression pattern. 2. Data presentation needs to be rearranged. It seems that authors used at least 3 different bladder cancer cells manipulating TAGLN expression. Throughout the article new cells keep popping up. 3. To propose TAGLN is suppressed in cancers, tissue localized expression of TAGLN needs to be presented (ie immunohistochemistry) since there are normal contributions of TAGLN from smooth muscles and fibroblasts. 4. Data presented throughout the manuscript is not consistent. In figure 2I, cytosolic TAGLN between mock cells and overexpressed cells shows not much difference for TAGLN expression (note that GAPDH is also a little high in overexpressed cells) while confocal shows dramatic expression difference. Also in Figure 3B, there is big expression difference even authors used whole cell extracts. Authors need to explain that. In figure 3, overexpression in HT cells leads EdU index from 47.16 to 39.24 while Sup Figure 1 shows Ki67 index from 43.69 to 25.93 using same cells. Different assays should not make this much difference. Minor issues: 1. While there is anti-TAGLN antibodies available, why authors had to use QRT-PCR for many experiments. 2. Most figures authors used are incorrectly manipulated, being stretched out horizontally. More importantly, many figures presented in this article needs to be updated (For example, figures like confocals in Figure3, invasion pictures in Figure 4, and Sup Figure 2). Figures themselves are not clear to take author’s point.

Author Response

For reviewer 2:

Thanks for your kindly support our manuscript. Grammatical and writing style errors in the original version have been corrected in green font by our colleague who is a native English speaker. We used red font to highlight the modifications in the revised manuscript to indicate where the changes we have made occur in the text. According to the reviewer recommendation, several modifications of this manuscript are listed as following: We agreed reviewer concerning the Figure 2I. We have replaced new results in the figure. The new figure showed clearly the higher expression of TAGLN in HT-TAGLN cells compared to HT-DNA cells; however, the extraction of cytosol and nucleus was not as clearly as previous figures The reason why we determined the expression of TAGLN after ectopic expression or knockdown TAGLN by using RT-qPCR and immunoblot assays is to confirm the expression of TAGLN at both transcriptional and translational levels. We agreed reviewer’s concerning the IHC assays to confirm the expression of the TAGLN in the bladder tissues. We also have concerned the IHC study in our early experimental design, and we, indeed, have done the preliminary study of IHC in the bladder tissue. However, as showed in our Figure 1. the HBdSMC and HBdSF cells express extreme high levels of TAGLN. Therefore, results of IHC assay only show the extensive strong positive results in the stromal cells which caused the signal for TAGLN in transitional epithelium cells are too weak to stain. If we wanted to show any signal showed in the epithelium cells need to over-stain the signal of TAGLN in stromal cells. Therefore, the extensive staining intense in stromal cells interrupted the results of signal of TAGLN in epithelium cells. The IHC study of bladder tissues would not be the good way to compare the TAGLN expression in epithelium cell in the normal and cancer tissues We agreed the reviewer’s concerning the results between EdU and Ki67 assays. In fact, this is our first study which used the EdU and Ki67 to measure the cell proliferation. Our results in Figure 3 and Figure supplementary 1 got similar pattern of proliferation from both assays although Ki67 assay has significantly different between groups than EdU assay. Since our study only one results using the Ki67; therefore, we set the results of Ki67 assay into the supplementary data. In fact, both assays are used for cell proliferation assay, there principle are quite different. The EdU can be incorporation into the newly synthesized DNA of replicating cells; therefore, the EdU assay is in measuring the S phase. The Ki67 protein, a cellular marker for cell proliferation, is present during all active phase of the cell cycle (G1, S, G2, and M). Therefore, Ki67 assay is considered to be a better marker for cell in growth fraction. We moved the data of Ki67 to supplementary data set. We agreed the reviewer’s concerning the signal pathways of p53 and PTEN on TAGLN gene. However, this study is the first study showed the biologic functions and regulatory mechanisms of TAGLN in bladder carcinoma cells. We found that TAGLN is upregulated by p53 and PTEN which was showed in Figure 6. We also approved the concept by ectopic overexpression or knockdown p53 in bladder carcinoma cells and confirmed the concept by using RT-qPCR, immunoblotting, and reporter assays. In the PTEN studies, we also approved the concept by ectopic overexpression and knockdown PTEN which alter the AKT phosphorylation and TAGLN expressions in bladder carcinoma cells. The precise signal pathways of p53 and PTEN on TAGLN gene expression will be our next issue. We agreed reviewer’s suggestion. The description of Figure 6 was reorganized and rewrote in the section of Results 2.7 as following:

L181-194: The immunoblot (Figure 6A) and RT-qPCR (Figure 6B) assays. confirmed that the overexpression of p53 induced TAGLN expression in HT1376 cells.  Further, the immunoblot assays  showed that camptothecin, a topoisomerase inhibitor, induced not only p53 but also TAGLN expressions, while p53-knockdown attenuated these effects in p53-wild type RT-4 cells (Figure 6C). Similar results found in the reporter assays indicated that transiently cotransfected-p53 expression vector induced the reporter activity of TAGLN reporter vector containing the 5’-flanking region of human TAGLN gene in HT1376 cells (Figure 6D). Further immunoblot assays revealed that ectopic overexpression of PTEN in T24 cells blocked AKT phosphorylation but enhanced TAGLN expression; whereas, PTEN knockdown in RT-4 cells reversed these effects (Figure 6E). Results of RT-qPCR assays showed that ectopic overexpression of PTEN induced TAGLN gene expression in T24 cells (Figure 6F), while PTEN-knockdown downregulated TAGLN expression in RT-4 cells (Figure 6G). Further reporter assays revealed that transiently cotransfected-PTEN expression vector upregulated the reporter activity of human TAGLN reporter vector in HT1376 cells (Figure 6H). Collectively, TAGLN is upregulated by p53 and PTEN in bladder carcinoma cells.

We agreed reviewer’s suggestion and reorganized the section of Conclusion as following.

L439-422: Our experiments provided evidence suggesting that TAGLN is a p53- and PTEN-downstream gene which attenuated cell proliferation and invasion in vitro and tumorigenesis in vivo. TAGLN seems to function as a tumor suppressor gene in bladder carcinoma cells.

We reorganized the figure to approve the quality of Figures.

Reviewer 3 Report

The manuscript from Ke-Hung Tsui and colleagues investigates transgelin in human bladder carcinoma cells.  A major problem is that the authors have not distinguished between the three different human transgelin genes (TAGLN, TAGLN2, and TAGLN3), which are located on different chromosomes and encode three different transgelin protein isoforms.  Online data sources (eg Human Protein Atlas) indicate that TAGLN (SM22) itself is restricted to smooth muscle/myoepithelial cells but is neither expressed by normal epithelium nor its derived cancers and cell lines.  By contrast, there is similar evidence that TAGLN2 is expressed in some epithelial cancers, including urothelial cancers.  Throughout the manuscript, the authors refer only to TAGLN, but it is essential that the authors provide evidence of the specificity of reagents they have used (including primer sequences and antibodies) in order to ensure there is no cross-reactivity in detection of different Transgelin gene transcripts or protein isoforms.  The authors have expressed TAGLN in bladder carcinoma cells, but it is perhaps therefore unsurprising that ectopic expression of non-tissue-relevant actin cytoskeletal regulatory protein has shown a negative influence on migratory/proiferative cell activity. This severely undermines the relevance of the study.

Author Response

For reviewer 3:

Thanks for your kindly support our manuscript. Grammatical and writing style errors in the original version have been corrected in green font by our colleague who is a native English speaker. We used red font to highlight the modifications in the revised manuscript to indicate where the changes we have made occur in the text. According to the reviewer recommendation, several modifications of this manuscript are listed as following: This study is the first study indicated that expression of TAGLN not only in bladder smooth muscle cells and fibroblast cells but also in primary bladder primary normal epithelial cells and bladder carcinoma cells. Reviewer concerned that our study did not compare the transglin-2 and the results may due to fault results since the data from Human Protein Atlas indicated that TAGLN Selective cytoplasmic expression in smooth muscle and myoepithelial cells. The results of dataset of Human Protein Atlas is based on IHC. As showed in our Figure 1. The HBdSMC and HBdSF cells express extreme high levels of TAGLN. Therefore, results of IHC assay only show the extensive strong positive results in the stromal cells which caused the signal for TAGLN in transitional epithelium cells are too weak to stain. If reviewer look clearly in Human Protein Atlas. It also showed the cell staining of TAGLN in Human Osteosarcoma U-2 OS cells (https://www.proteinatlas.org/ENSG00000149591-TAGLN/antibody); therefore, TAGLN should not only express in smooth muscle and myoepithelial cells.. We did not understand why Human Protein atlas indicated that point since TAGLN has found in several cell types and also as the tumor marker in several cancer as we described in the section of Introduction. Our study using the commercial available antibody and probe specific for TAGLN in immunblotting, immunostaining, RT-qPCR assays should clearly the expression of TAGLN in bladder carcinoma cells. Recent study by Chen et al., (EBioMedicine 2019, doi: 10.1016/j.ebiom.2019.08.012.  also demonstrated the expression of TAGLN in bladder carcinoma cells. In generally concept, the transgelin-2 has contradictory to the role of TAGLN in the cancer development (Dvorakove et al., Exp Rev Proteomic, 2014). In this study, we focused on the TAGLN and did not repeat the experiment on transgelin-2 since the transgelin-2 is well-known as the oncogene of bladder cancer in vitro and in vivo (Yoshino et al., 2014; Zhang et al., 2018). We understand the concerns from the reviewer; therefore, we added several paragraph to explain the results in the section of Discussion as following:

L232-234: Meanwhile, a recent study showed that TAGLN was expressed in bladder carcinoma T24 and SW780 cells at the transcription and translation levels [25].

L239-245: The family of transgelins consists of three homologs (TAGLN, transgelin-2, and transgelin-3) in human tissues. Prior study has suggested that TAGLN and trangelin-2 acted as cancer biomarkers and differentially expressed in the tumor and stroma cells [27]. Further studies implied that transgelin-2 has the potential oncogenic function in bladder cancer in vitro and in vivo [28,29]. Our study presents the tumor suppressor characteristics of TAGLN in bladder cancer in vitro and in vivo is in agreed to the concept of that the role of transgelin-2 in the tumor development might be contradictory to the role of TAGLN [27].

We added 4 new references as following:

L515-514:

Chen, Z.; He, S.; Zhan, Y.; He, A.; Fang, D.; Gong, Y.; Li, X.; Zhou, L. TGF-b-induced transgelin promotes bladder cancer metastasis by upregulating epithelial-mesenchymal transition and invadopodia formation. EBioMedicine 2019, doi: 10.1016/j.ebiom.2019.08.012. 

L517-523:

Dvorakova, M.; Nenutil, R.; Bouchal, P. Transgelins, cytoskeletal proteins implicated in different aspects of cancer development. Expert. Rev. Proteomics 2014, 11, 149-65. 

Yoshino, H.; Chiyomaru, T.; Enokida, H.; Kawakami, K.; Tatarano, S.; Nishiyama, K.; Nohata, N.; Seki, N.; Nakagawa, M. The tumour-suppressive function of miR-1 and miR-133a targeting TAGLN2 in bladder cancer. Br. J. Cancer 2011, 104, 808-818.

Zhang, H.; Jiang, M.; Liu, Q.; Han, Z.; Zhao, Y.; Ji, S. miR-145-5p inhibits the proliferation and migration of bladder cancer cells by targeting TAGLN2. Oncol. Lett. 2018,16, 6355-6360.

Round 2

Reviewer 1 Report

The manuscript has been modified extensively and detailed have been added. There are minor grammatical errors, that need to be fixed before publishing.

Reviewer 2 Report

Revised article is much improved. Only one thing to check is that authors said manuscript has been edited by native English speker but I still see some faults, such as "in agreed to". Is that right term? Sincerely,

Reviewer 3 Report

The authors have performed an in depth multifaceted study, but the authors’ contention that TAGLN represents a tumor suppressor gene downregulated in bladder carcinoma, is still unsupported by a lack of unequivocal evidence that TAGLN is expressed by normal bladder epithelial cells; the work is therefore unfortunately flawed. 

TAGLN is a transcript detected in many heterotypic tissues/organs due to its expression by cells of mesenchymal derivation.  From sources such as The Protein Atlas there is no compelling evidence that it is a gene expressed by normal epithelial cells, such as urothelium.  For this reason, the level of evidence supplied by the authors needs to be watertight, which it is currently not.

There are issues over experimental controls and particularly a lack of evidence supporting the specificity of cells and antibody reagents used in their study. 

Controls shown from RTPCR reactions do not contain adequate controls (for example RT- controls and template only controls to eliminate the possibility of priming from DNA or other contaminants). The primer sequences used for RTPCR are not included.  The human normal primary bladder epithelial cells (HBdEC) used show amplification of aSMA indicating that the preparation contains at least some stromal cell contamination. The manuscript does not indicate the source or nature of the anti-transgelin antibodies used – only that it was a rabbit antiserum (P12 line 366). This is inadequate information, particularly as many polyclonal antibodies show cross-reactivity. Just because it is commercially sourced does not make it a good reagent and better controls are required to demonstrate specificity.  Why is transgelin not shown on blot 1C?